# MicroRNA Milk Exosomes: From Cellular Regulator to Genomic Marker

**DOI:** 10.3390/ani10071126

**Published:** 2020-07-02

**Authors:** Michela Cintio, Giulia Polacchini, Elisa Scarsella, Tommaso Montanari, Bruno Stefanon, Monica Colitti

**Affiliations:** Department of Agriculture, Food, Environmental and Animal Science, University of Udine, 33100 Udine, Italy; michela.cintio@uniud.it (M.C.); giulia.polacchini@uniud.it (G.P.); scarsella.elisa@spes.uniud.it (E.S.); tommaso.montanari@uniud.it (T.M.); monica.colitti@uniud.it (M.C.)

**Keywords:** bovine milk, exosomes, miRNA, ruminants, genome, mastitis

## Abstract

**Simple Summary:**

Bovine milk contains proteins, minerals (e.g., calcium), vitamins (A, B, D, E and K), bioactive compounds and signaling molecules, as short non-coding RNA, either floating or contained in exosomes. Milk compositions and the milk-derived exosomes’ (EXO) cargo varies in relation to the health status of the mammary gland and the lactation stage. Stress, immune-depression and mastitis generate a differential microRNA (miRNA) expression, and consequently a modulation of the local immune cell response. This review aims to discuss the possible application of EXO cargo as a biomarker of animal resilience, and as a phenotype for genomic studies.

**Abstract:**

Recent advances in ruminants’ milk-derived exosomes (EXO) have indicated a role of microRNAs (miRNAs) in cell-to-cell communication in dairy ruminants. The miRNAs EXO retain peculiar mechanisms of uptake from recipient cells, which enables the selective delivery of cargos, with a specific regulation of target genes. Although many studies have been published on the miRNAs contained in milk, less information is available on the role of miRNAs EXO, which are considered stable over time and resistant to digestion and milk processing. Several miRNAs EXO have been implicated in the cellular signaling pathway, as in the regulation of immune response. Moreover, they exert epigenetic control, as extenuating the expression of DNA methyltransferase 1. However, the study of miRNAs EXO is still challenging due to the difficulty of isolating EXO. In fact, there are not agreed protocols, and different methods, often time-consuming, are used, making it difficult to routinely process a large number of samples. The regulation of cell functions in mammary glands by miRNAs EXO, and their applications as genomic markers in livestock, is presented.

## 1. Introduction

Prokaryote and eukaryote cells release extracellular vesicles (EVs) as part of their normal physiology, and as a result of acquired abnormalities [1]. The cells of multicellular organisms use different ways to communicate, including EVs, which play a unique role [2]. A particularly interesting subtype of EVs is exosomes, the research into which is rapidly growing.

Exosomes are membrane-bound vesicles with a size range of about 40 to 160 nm in diameter (average about 100 nm), and are round or cup-shaped when observed under transmission electron microscope [1]. Indeed, exosomes are generated in a process that involves double invagination of the plasma membrane. The inward budding of late endosomes produces multivesicular bodies (MVBs), containing intraluminal vesicles that can fuse with the cell membrane and are released into the extracellular environment. Once released into extracellular space, these nanovesicles are termed exosomes [3,4]. Exosome secretion requires the sequential assembly of four endosomal sorting complex required for transport (ESCRT) complexes on the endosomal membrane.

A clear classification of EVs is difficult [5,6], but exosomes have some distinctive features, such as enzymatic and transport differences. These features distinguish exosomes in milk, here referred to as EXO, from milk fat globule membranes, which also have a plasma membrane origin [7].

The final content of exosomes is produced by subsequent interactions with other intracellular vesicles and organelles, contributing to their diversity. Depending on the cellular origin, exosomes can contain many cell constituents, including nucleic acids, such as messanger RNA (mRNA), microRNAs (miRNAs), ribosomal RNA (rRNA), long noncoding RNA (lncRNA), transfer RNA (tRNA), as well as, variably, DNA, lipids, metabolites and proteins (Figure 1). The differences in exosome contents and their involvements in different biological processes represent a fingerprint of the donor cell [8,9]. Then, the different contents of exosomes can be delivered to a recipient cell, whose biological response can be effectively altered [1]. In addition, exosomes secreted by cells are found in various biological fluids such as blood, plasma, tears, semen, saliva, urine, cerebrospinal fluid, epididymal fluid, amniotic fluid, tumor ascites, bronchoalveolar lavage fluid, synovial fluid and milk [2,10,11]. As result, exosomes are used as biomarkers for the diagnosis of various diseases [1,12,13].

Exosomes, through their cargos, play significant roles in many biological processes, such as angiogenesis, coagulation, cell proliferation and differentiation, apoptosis, antigen presentation and inflammation [10]. Moreover, exosome cargos influence physiological and pathological processes in various diseases, including cancer, neurodegenerative diseases, infections and autoimmune diseases. Concerning the immune system, exosome cargos allow the exchange of antigens, or major histocompatibility complex–peptide complexes, between antigen-bearing cells and antigen-presenting cells [3]. Exosomes can also induce tolerance against allergens, the inhibition and activation of natural killer cells, and the differentiation, proliferation and apoptosis of T regulatory cells [8]. In cancer, they may contain molecules leading to adverse effects on recipient cells. In fact, exosomes derived from tumor cells have been associated with accelerated tumor growth [14,15] and invasiveness [16]. Indeed, exosomes containing tumor-specific RNAs and proteins are used as biomarkers for cancer diagnosis [10] and, by proteomic analysis, they indicate whether the cell of origin was epithelial-like or mesenchymal-like [1,17].

Exosomes can contain higher concentrations of miRNAs compared to the origin cells, indicative of selective wrapping [18]. miRNAs are a class of single-stranded noncoding RNAs about 22 nucleotides long, which induce post-transcriptional gene silencing by binding the 3′UTR (untranslated region) of target genes. They are generated in the cell cytoplasm, by the ribonuclease III Dicer, from nuclear precursor transcripts. Subsequently, they cause degradation mediated by proteins of the Argonaute family (Ago), or the translational repression of totally or partially complementary mRNAs [19]. Short sequence motifs of miRNAs controlling their loading into exosomes have been described [20]. These short non-coding RNAs can regulate up to 60% of gene expression at the post-transcriptional level [21,22]. Thereby, they regulate gene expression, and the usual consequence is the downregulation of protein expression [23]. In this way, miRNAs control a wide range of cellular functions, such as cell differentiation, organogenesis [24], proliferation and cell death, and regulate several aspects of health and diseases [25,26], fat storage, hematopoiesis and immunity [27]. Authors have reported that miRNAs constitute the majority of the RNA contained in these vesicles [2,28,29]. However, the stoichiometry analysis of miRNAs in exosomes, performed by Chevillet et al. [30], indicated that the copy numbers of these molecules are very low, and a reconsideration of the mechanism of exosome-mediated miRNAs communication deserves further investigation.

Milk contains many compounds [31,32,33], and is also the primary source of nutrition for newborn mammals; in fact, it is the only food that ensures infant development and health in short-time and long-time ways [34]. Breastfeeding is associated with significant health benefits, including protection against infection, obesity, diabetes and cancer [35,36]. Moreover, breast milk plays an important role in the development of a child’s immune system [37], and helps the infant’s intestine to develop, which becomes able to absorb milk and is prepared for weaning [38]. Milk from cattle [39,40], goats [41], humans [42] and rodents [43] also includes different biologically active EXO that explicate their functions in a manner dependent on their miRNA contents. Interestingly, the most abundant miRNAs EXO are found among humans, swine, cows and pandas [44]. Golan-Gerstl et al. [41] claim that 95% of the miRNAs expressed in human milk is also expressed in bovine and goat milk, and that bovine milk pasteurization does not seem to destroy miRNAs. Moreover, 24 validated targets were shared among conserved miRNAs, and these targets were involved in immunomodulatory functions [44]. miRNAs EXO can be transferred from mother to infant because they are packed into EXO, which protects them from degradation and from the infant gastrointestinal environment [45,46]; in fact, miRNAs EXO are stable in very acidic conditions (pH 1) [47]. Indeed, miRNAs EXO can cross the intestinal barrier, and are able to modify the gene expression of recipient cells [48,49]. EXO facilitate the absorption of miRNAs into the intestine through endocytosis, influencing the immune system, as the intestine is one of the major immune organs [38]. The incorporation of cow and goat milk EXO into human colon cancer cells [50], intestinal cells [40], kidney cells and peripheral blood mononuclear cells [41] has been reported. Hence, EXO make miRNAs stable, bioaccessible, bioavailable and bioactive [27].

Several studies have shown that over 60% of miRNAs EXO are related to immune regulation [51]. Human breast milk is rich in T cell-regulating miRNAs [52], and in miR-181 and miR-155, which induce B-cell differentiation [53]. This can explain why infants that are fed on breast milk, compared to infants fed on infant formula, have a reduced incidence of digestive problems, and can be more protected against gastrointestinal and respiratory infections [38].

Similar to the case with milk, bovine colostrum is rich in the common and most abundant proteins (caseins, β-lactoglobulin and α-lactalbumin), but also in monocyte differentiation antigen CD14 (CD14), glycosylation-dependent cell adhesion molecule 1 (GLYCAM1), xanthine dehydrogenase/oxidase (XDH/XO), lactadherin (MFGE8) and clusterin (CLU) [54]. Some of these proteins are implied in the modulation of the immune response of the offspring. Furthermore, colostrum contains large amounts of EXO enriched in proteins and in miRNAs (e.g., miR-223) responsible for immunity regulation [55,56]. It is likely that the miRNAs EXO of colostrum, and those delivered to the calf with milk during weaning, play an essential role in organism development and immune response [57]. Whether this will influence animal wellbeing later during maturity deserves further investigation.

Because EXO show very intriguing aspects, their isolation and characterization deserves particular attention, since agreed protocols are lacking and different methods, often time-consuming, are used, limiting the possibility of applying them to a large number of samples.

## 2. Isolation Methodologies

Isolation of exosomes is the first challenging step to face in biomedical research, or in any other application field. Since the exosomes have very small dimensions, their isolation turns out to be a difficult process. The choice of the most appropriate method depends on the chemical and physical properties of the biological matrix, which can be an obstacle to the extraction of the nanovesicles. Bovine milk contains caseins, and therefore usual techniques for isolating EXO must be modified accordingly (Table 1).

### 2.1. Ultracentrifugation

Ultracentrifugation (UC) is the conventional and most utilized method, due to its applicability in pelleting lipoproteins, extravesicular protein complexes, aggregates and other contaminants, but it is not suitable for EXO isolation from clinical samples, because it is time-consuming, labor-intensive, requires costly instrumentations and includes multiple centrifugation steps [58].

According to Li et al. [59], two types of UC methods exist: analytical and preparative. The first one is used to understand the physicochemical properties of materials and the molecular interactions of polymeric materials, while the latter aims to fractionate biological elements (microorganisms, viruses, organelles and EVs). In particular, differential UC and density gradient UC are the two preparative election techniques for isolating EXO from milk. The second one gives the purest EXO population when compared to the other type of UC, according to Gurunathan et al. [60]. This method was also supported by Hata et al. [61], who obtained, through sucrose density-gradient ultracentrifugation, a suitable recovery of microvesicles from bovine milk to proceed with RNA isolation.

Yamada et al. [62] stated that density gradient UC is the most appropriate method for isolating high-quality EXO with intact morphology. Therefore, this method can be used to examine the relation of the exosomal proteins to the physiological or pathological status of the host, without any involvement or contamination of the other free proteins in milk. On the contrary, Vaswani et al. [63] obtained a population of nanovesicles from fresh bovine milk through differential UC, which they then enriched through size exclusion chromatography (SEC), or filtered through a 0.22-µm membrane [48], before the execution of the downstream application.

### 2.2. Isoelectric Precipitation

Milk contains many proteins (β-lactoglobulin, serum albumin, casein, etc.) that hinder EXO isolation and purification. In particular, the large quantity of caseins, which are the major milk proteins, accounting for more than 80% of total bovine milk proteins (in contrast to 35% of human breast milk proteins [64]), hinders the direct use of conventional methods for the isolation of bovine milk EXO. As such, some scientists contrived a specific method for removing caseins through isoelectric precipitation (IP). Yamauchi et al. [65] reported that the acidification of defatted bovine milk to pH 4.6, by adding HCl 6N, facilitates the precipitation of proteins, and improves the following steps of EXO isolation. Rahman et al. [66] compared two different acidification protocols by the addition of HCl or acetic acid to defatted milk. It was shown that neither acid condition alters EXO cargo, but both produce a rough surface and partial degradation of the EXO membrane proteins. Even Somiya et al. [67], before acidifying purchased defatted milk by adding acetic acid 1N, pre-warmed it, and then ultrafiltered the sample. However, even there is still no univocal protocol for isolating the EXO from milk, acidification seems to be the most suitable step for removing caseins (Table 2). To improve the isolation of EXO from raw milk, recently, Li et al. [68] suggested freezing the purified whey at −80 °C prior to the ultracentrifugation, in order to eliminate cell debris and apoptotic bodies, which can hinder the purity of the final pellet. The protocol that we have used for EXO isolation is reported in Table 3. This method allows us to obtain a satisfactory yield of EXO with high purity (Figure 2 and Figure 3).

Care must be taken in isolating milk EXO, but also in the downstream applications. Usually, the pellet has to be suspended in a buffer (e.g., Phosphate-Buffered Saline (PBS)0.1M) when EXO characterization is performed, but if miRNAs-derived EXO isolation is required, resuspension in a lysis buffer is necessary [69].

### 2.3. Immuno-Affinity Purification

Exosomes contain, and expose on their surface, a lot of characteristic proteins. Immunoaffinity techniques exploit the interaction between exosomal proteins (antigen) and their antibodies [68] to obtain a final pure concentration. Exosomal samples derived from different biofluids (e.g., urine, saliva, milk) contain a mix of other biological components and proteins, which increases the difficulty of obtaining a pure population of EXO. To overcome this issue, IP techniques take advantage of the exosome-specific antibodies, such as anti-CD9, anti-CD63 and anti-CD81.

Furthermore, magnetic beads, covalently coated with streptavidin, allow a more stable bond between antigens and any biotinylated capture antibody. In a comparison study between different methods, starting from cell culture media, Greening et al. [70] determined that the immuno-affinity magnetic technique, enriched for EXO and exosome-associated proteins, is at least twofold more efficient than the other methods used. Immuno-affinity methods with magnetic beads seem to be even more efficient, increasing capture and purity of the EXO from human serum samples, as reported in Yoo et al. [71].

### 2.4. Microfluidics-Based Isolation Techniques

In addition to the usual isolation methods, new techniques have been developed to deal with the problems of time-consuming steps, expensive lab machinery and high purity final pellets. These systems are based on the common separation determinants (size, density, immuno-affinity), but with the addition of innovative sorting mechanisms, e.g. acoustic, electrophoretic and electromagnetic manipulation, or viscoelastic flow. According to their procedures, these techniques can be divided into two categories: trapping by immune-affinity, sieving (e.g., nanoporous membranes), and trapping EXO on porous structures (e.g., nanowire-on-micropillars) [72].

Chen et al. [39] developed the first immune affinity microfluidic device for the capture of EXO, based on the bond of CD63, from serum samples. Kanwar et al. [73] obtained a pure population of EXO by using an Exo-chip platform, which also allowed EXO quantification by the fluorescent assay method. These methods are quicker than ultracentrifugation, and require low volumes of samples and reagents.

Im et al. [74] designed an on-chip nano plasmonic EXO sensor (nPLEX), in which surface plasmon resonance (SPR, a technique based on the oscillations of electrons at an interface stimulated by incident light) is used through nanohole arrays patterned on a metal film.

A collection of EXO was obtained by directly sieving whole blood through a membrane as a size-exclusion filter, but this method offered a low recovery of EVs [75].

Regarding the trapping of EXO on porous structures, Wang et al. [76] devised a ciliated micropillar plate that formed a microporous silicon nano-wire, which was able to selectively trap particles in the range of 40–100 nm, facilitating the selective collection of intact phospholipidic “EXO-like” vesicles. Unfortunately, the device was not validated with clinical samples, and no analysis of cargo protein or RNA was done.

## 3. Characterization of EXO

All body fluids produced by animal organisms contain EVs coming from different tissues. When analyzing exosomes isolated from body fluids, characterization is a crucial phase in discriminating exosomes from other families of EVs. An accurate method of characterization is necessary to clearly understand the roles of exosomes and other EVs in physiology and pathology [77]. So far, there is no gold standard technique for characterizing exosomes via all their physical and bio-molecular properties. The choice of the characterization protocol relies on sample type, isolation method and the application of isolated exososmes. In the case of bovines, different exosomes-containing body fluids have been deeply analyzed, including colostrum and milk [78,79,80,81,82], serum and plasma [78,83,84,85,86,87,88], urine [78,89], and saliva [78].Through comparative studies of different bovine biological fluids, a recent study reported that plasma and milk collected from cows have a significantly higher abundance of EXO, in comparison to their urine and saliva [78].

### 3.1. Size Characterization of EXO

One method for characterizing and discriminating EXO from EVs is flow cytometry (FC) [90,91]. In FC, a cell/vesicle suspension is forced through a laser beam, which scatters when it is crossed by a cell/vesicle, producing a fluorescent signal detected by a series of detectors. Despite the fact that this method is widely preferred, conventional FC protocols are challenged by the small size of EXO, due to the detection limit of flow cytometers, which is around 100 nm. This has resulted in the implementation of EXO-specific protocols, as described in several papers [91,92,93,94].

Nanoparticle tracking analysis (NTA) identifies, by illuminating with a laser beam, EVs in suspension, and detects their position with a dark-field microscope [95]. Although particles smaller than 200 nm are below the canonical light microscopy resolution limit, laser scattered light produced by suspended vesicles can be seen by the image sensor of a camera, allowing visualization of the vesicle’s position [96]. Since vesicles in suspension are characterized by Brownian motion, NTA tracks their movement in real time, and measures their mean velocity, which, being dependent on their size, can produce an absolute calculation of their diameter [97]. Most NTA instruments are now equipped with fluorescence detection, coupling NTA, with the possibility of phenotyping vesicles and EXO that react with fluorescently labeled antibodies [77], with a lower risk of artefacts [98]. NTA is greatly applicable in the rapid assessment of EXO size, refractive index and concentration [96]. NTA detects EXO in the 50–250 nm range (Figure 2), which overcomes the detection limit of FC [77,96,99].

### 3.2. Protein- and Microscopy-Based Characterization

Marker-based EXO characterization is an experimental procedure based on the detection of EXO-specific proteins through Western blotting and/or electron microscopy-coupled immunoreaction. The protein characterization of EVs from different origins is analyzed by sodium dodecyl sulphate—polyacrylamide gel electrophoresis (SDS-PAGE) separation, followed by protein staining, immunoblotting or proteomic analysis [100]. Immunogold labeling in transmission electron microscopy (TEM) characterizes the biochemical properties of EXO surface proteins (Figure 3). All steps are carried out on EXO directly bound to formvar carbon-coated nickel grids [90,101].

EXO biomarkers are listed on www.exocarta.org [12] according to how many times they have been identified. CD9 is the most identified tetraspanin in EXO. It is involved in cell motility, adhesion and fusion [102], and its exosomal enrichment has been observed in EXO [103,104].The tetraspanin CD63 is also one of the most used markers, since it is particularly enriched in the intraluminal membrane of multivesicular bodies [105]. The detection of CD63 by Western blotting and immunogold has been extensively used in bovine EXO characterization [65,101,103]. Further, the tetraspanin CD81 is a selective exosomal biomarker in the milk of woman and cows [106]. The heat shock cognate 70-kDa protein (HSC70) and heat shock 60-kDa protein (HSP60) are molecular chaperones whose enrichment has been observed in the EXO generated by multiple tissues. HSC70 has been shown to be enriched in bovine EXO following immunogold analysis [101]. HSP60 is normally expressed in cytosol and mitochondria, but it can also be uptaken in EXO. A role in cell-to-cell communication has been attributed to EXO enriched in HSP60 and HSP60 since they are a Toll-like receptor ligands [107]. HSP60-enriched EXO have also been identified in bovine milk, making this chaperone another interesting biomarker for EXO characterization [108]. The minimum information required for the study of extracellular vesicles is reported by Théry et al. [6].

With respect to the antibody-dependent microscopical characterization of EXO, atomic force microscopy (AFM) presents great advantages. AFM’s principle relies on the interaction between the probing tip of the instrument and the surface of the sample, which is probed by a cantilever ending in a sharp tip. When the cantilever is very close to the sample, the atomic forces of the two interact, and this produces a deflection of the cantilever, which is then recorded by a laser-photodiode system [109]. Sample preparation and image acquisition are very simple, since atomic force microscopes work under ambient conditions, and EVs can be imaged and do not require further downstream isolation procedures [110,111]. Moreover, the EVs binding specific antibodies can be used to detect the presence of specific proteins with a better resolution than immunogold labeling. EXO analysis is simplified by the fact that AFM reaches a sub-nm resolution, at which it is possible to appreciate the morphology of a single EXO, and to count EXO in a label-free and unbiased fashion. The high resolution power of AFM also allows the quali–quantitative assessment of surface exosomal molecules [112].

## 4. Roles of miRNA EXO in Metabolism and Health

Numerous studies have been reported on milk-derived EXO, of human and laboratory animal origin, with both in vitro and in vivo approaches, but less information is available on the EXO in dairy ruminants. In humans, the molecular composition of milk EXO depends on the physiological status of the mother [2], and largely on her lifestyle, lactation stage and contact with allergens [113].

Since a strong relationship between the lactation curve and the plethora of pathways involved in the mammary gland exists, the specific cargo of the EXO can change during the different phases of the lactation cycle. Many of the miRNAs EXO originate from the mammary epithelium [114,115], and the modulation of miRNAs expression in mammary cells impacts the milk profile composition. It has been found that miR-221 is expressed differentially during mammary development, and regulates the lipid metabolism of the mammary cells of mice [116]. It has recently been stated that the miRNAs profile in goat milk exosomes changes during lactation, affecting milk fatty acids (FA) content through transcriptome modifications in the mammary epithelial cells [117,118]. For instance, miR-27a and miR-183 have been shown to promote the content of unsaturated FAs and medium chain FAs [117,118]. The likely mechanism underlying the variations in the FA profile of goat milk is the silencing of key genes involved in lipid metabolism. Lin et al. [117] stated that miR-27a downregulates stearoyl-CoA desaturase (SCD), while Chen et al. [118] found that the silencing action that miR-183 exerts on the MST1 gene leads to the upregulation of adipose triglyceride lipase (ATGL), carnitine palmitoyltransferase 1 (CPT1) and SCD [119].

The EXO protein and miRNA content play a role in modulating the inflammatory response and the development of the host [27,120,121,122,123]. Of note, miRNAs with immune-related activities are abundant in breast milk EXO [124]. Little information is available for cattle, but a change in the composition of the proteins [125] and nucleic acids [126] of EXO occurs during the inflammatory response to pathogens in the mammary gland. Stressful conditions can modify miRNAs derived from milk EXO as well. During the relocation of dairy cows in early lactation, 15 miRNAs were found to be differentially expressed [101]. Therefore, milk miRNA profiling could be useful for the early diagnosis of infections, and for monitoring the immune status of dairy cows [27]. According to Munagala et al. [48], a large copy number of genes and miRNAs, related to bovine mammary glands and immune function, have been found. Furthermore, miRNAs from EXO have been demonstrated to play a role in mouse thymic regulatory T (Treg) cell differentiation [38,51,127,128]. In mouse models, EXO have anti-inflammatory power, and further prevent and decrease the symptoms of rheumatoid arthritis [129]. Moreover, the immunomodulatory activity of mice, after oral administration of EXO, has exhibited an increase in M1 macrophage polarization and inflammatory cytokine secretion (IL-6, TNFα, IL12 and IL23) [130].

Most miRNAs derived from milk EXO are involved in immune functions, such as miR-148 and let-7a. In particular, miR-148 appears to be a negative regulator of the innate immune response, and of the antigenic presentation function in mouse dendritic cells [131]. Furthermore, miR-148a is up-regulated during the cell differentiation [132], targeting DNA methyltransferase 3b [133] and DNA methyltransferase 1 [134] genes in humans. Both genes are involved in tumor development, proliferation and metastasis, and are down-regulated if miR-148a is up-regulated, demonstrating that this miRNA has a tumor-suppressing effect. This feature has raised concerns about the impact of recurrent milk consumption on the epigenetic regulation of the human genome [135,136]. Of note, miR-148a is the one of the most highly expressed miRNAs in bovine and goat milk, before and after pasteurization [41]. Let-7a inhibits Th17 cell differentiation through the downregulation of IL-6 secretion [137]. Interestingly, these miRNAs and others that are mostly expressed in milk EXO, are highly conserved among species [27] and, considering their immune regulatory role, can be candidates of mastitis biomarkers. The early diagnosis and effective treatment of cow mastitis can guarantee a healthy growth of dairy cattle and a qualified production of dairy products, contributing to reducing and/or improving the use of antibiotics on farms [138,139]. In the milk from mammary glands naturally infected by *Staphylococcus aureus*, another three miRNAs (miR-378, bta-miR-185 and miR-146b) were found to be significantly up-regulated [140]. The target genes of bta-miR-378 and miR-185 have been validated, and their role has been associated to health parameters [80].

## 5. miRNA EXO as a Genomic Markers

The RumimiR database curated by Bourdon et al. [140] contains a collection of miRNAs from three species of dairy ruminants (cattle, goat and sheep), obtained from different tissues and animal statuses. The authors reported miRNAs differences in the expression level not only between species, but also among breeds. Variations of miRNomes among breeds are recognized as expression quantitative trait loci (miR-e-QTL) [141], which paves the way for their use in breeding programs for polygenic complex traits. In turn, miRNAs contain genomic information that can assist researchers to discover gene functions and to provide a better understanding the regulatory mechanisms of gene expression [142].

Focusing on ruminants’ mammary glands, several studies have underpinned the participation of miRNAs in the regulation of lactation signaling pathways, physiological processes, and health [143] and milk traits [144,145].

In a comparison between Montbéliarde and Holstein lactating cows, Billa et al. [146] found 11 up-regulated and 11 down-regulated miRNAs in the mammary gland tissue, suggesting that their expression level is related to the milk, protein, lactose and fat yields being higher in Holstein than in Montbéliarde cows. The six most differentially expressed miRNAs were miR-100, miR-146b, miR-186, miR-30e-5p, miR-25 and miR-16a. In particular, miR-100 and miR-146b target mammary metabolism through the mTOR signaling pathway, and their differential expression has been related to milk yield. Furthermore, miR-100, together with miR-186, is also related to epithelial-to-mesenchymal transition, controlling the remodeling and development of mammary glands. Again, miR-100, together with miR-30e-5p, miR-25 and miR-16a, targets lipid metabolism by promoting milk fat synthesis and miR-186 regulates glucose uptake. Indeed, Le Guillou et al. [147] compared the milk miRNomes of Normande and Holstein cows, which differed in milk yield and composition, finding 182 miRNAs, differentially regulated, that are involved in lipid metabolism and mammary structure.

Since miRNAs EXO have a role in cell-to-cell communications, their association with complex traits would indicate that they could be useful as genomic markers, and not as just markers of mammary gland function.

## 6. Conclusions

Exosomes are a distinctive cell-to-cell communication mechanism, which can deliver molecules able to influence, in multiple ways, the biological function of the recipient cell. Among these molecules, miRNAs EXO represent an intriguing regulatory system, involved in the biology of the mammary glands, specifically with regard to immune responses and metabolic processes. Whether the miRNome EXO can be used as a genetic marker in lactating ruminants is still unresolved, and needs to be deeply investigated. Moreover, efforts are required to define standardized and agreed-upon methodological procedures for EXO isolation, in order to gain a deeper insight into their possible application in livestock production sciences.

## Figures and Tables

**Figure 1 animals-10-01126-f001:**
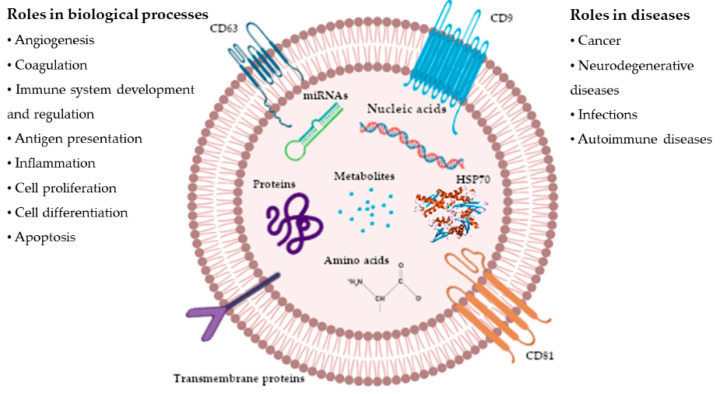
Simplified representation of the roles and contents of exosomes. Exosomes are extracellular nanovesicles generated by all cells, and they carry nucleic acids, proteins, lipids and metabolites. They are mediators of near- and long-distance intercellular communication in health and disease (modified from [1]).

**Figure 2 animals-10-01126-f002:**
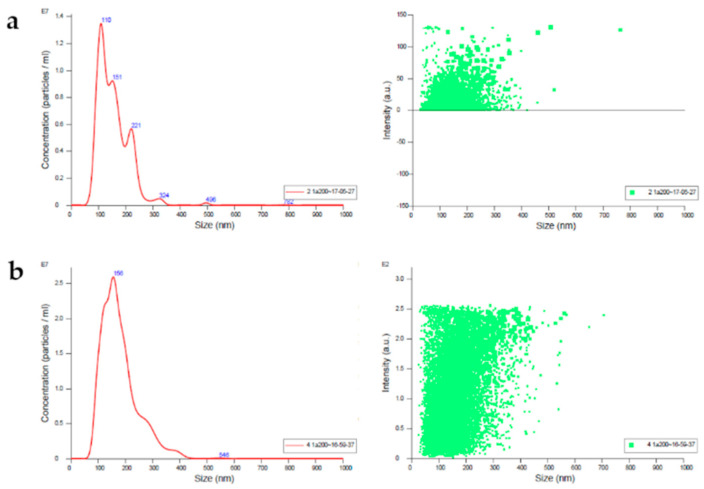
Quantification and size distribution of isoelectric precipitated and isolated milk exosomes, recorded through Nanosight (Malvern Panalytical, Malvern, UK) under light scatter mode. Red line indicates the signals recorded and depicts the extracellular vesicles (EVs) distribution in the sample, and the green spots indicate the intensity of the signals in an arbitrary unit (a.u.). (**a**) The sample has a concentration of 3.32 × 10^9^ particles/mL and denotes different EVs population, with peaks at 110, 151, 221, 324, 496 and 782 nm. The mean size is 178.4 nm and the mode is 155.4 nm; (**b**) The sample has a concentration of 1.34 × 10^9^ particles/mL and denotes only an EVs population with a peak at 156 nm, and with an irrelevant peak at 546 nm. The mean size is 154.2 nm and the mode is 109.3 nm.

**Figure 3 animals-10-01126-f003:**
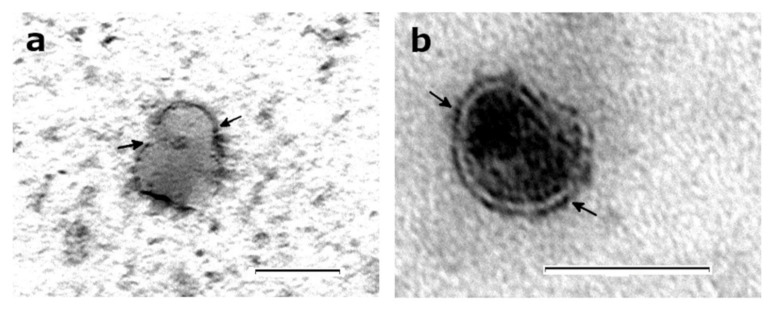
Exosomes were isolated from bovine milk according to the proposed method, and stained with antibodies (**a**) CD63 and (**b**) HSC70, coupled with a 10-nm gold particle. Images were taken using magnification 25,000× [(**a**), scale bar 200 nm] and 92,000× [(**b**), scale bar 100 nm].

**Table 1 animals-10-01126-t001:** Types of isolation technique used for milk exosomes (EXO).

Basic Technique	Method
Ultracentrifugation	Differential centrifugation
	Density gradient ultracentrifugation
	Isoelectric Precipitation
Microfluidics	Size exclusion (also magnetics activated)
Immunological method	Enzyme-linked Immunosorbent Assay (ELISA)

**Table 2 animals-10-01126-t002:** Comparison of main steps for exosome isolation from milk with ultracentrifugation, with or without isoelectric precipitation.

Steps	[67]		[65]			[66]	
	AA/UC Method	C/UC Method	UC Method	IP Method	AA Method	UC Method	IP Method
Centrifugation to defat milk	X ^1^	-	2000 *g* at 4 °C, 20 min		2000 *g* at 4 °C, 20 min	2000 *g* at 4 °C, 20 min	2000 *g* at 4 °C, 20 min
Distilled water addition	-	-	-	1:1	-	-	1:1
Warming	10 min at 37 °C	-	-	10 min at 37 °C	10 min at 37 °C	-	10 min at 37 °C
Casein precipitation							
Acidification	Milk/Acetic acid (1:1), 5 min RT	-	-	HCl 6N to adjust pH 4.6	Milk/Acetic acid (1:1), 5 min RT	-	HCl 6N to adjust pH 4.6
Centrifugation	10,000 *g* at 4 °C, 10 min	-	-	5000 *g* RT, 20 min	5000 *g* RT, 20 min	-	5100 *g* RT, 20 min
Filtration	0.22 μm	-	-	1.0, 0.45, 0.2 μm	1.0, 0.45, 0.2 μm	-	1.0, 0.45, 0.2 μm
Ultracentrifugation	210,000 *g* at 4 °C, 70 min	13,000 *g* at 4 °C, 30 min;	12,000 *g* at 4 °C, 60 min	-	-	12,000 *g* at 4 °C, 60 min;	-
		100,000 *g* at 4 °C, 60min;	35,000 *g* at 4 °C, 60 min			35,000 *g* at 4 °C, 60 min;	
		130,000 *g* at 4 °C, 60 min	70,000 *g* at 4 °C, 3 h			75,000 *g* at 4 °C, 3 h	
Filtration		-	1.0, 0.45, 0.2 μm	-	-	1.0, 0.45, 0.2 μm	-
Pellet resuspension with Phosphate-Buffered Saline (PBS) and ultracentrifugation	210,000 *g* at 4 °C, 70 min						

X ^1^ Defatted purchased milk; AA/UC = Acetic acid/ultracentrifugation; C/UC = Centrifugation/ultracentrifugation; UC = Ultracentrifugation; IP = Isoelectric precipitation; AA = Acetic acid; RT = room temperature.

**Table 3 animals-10-01126-t003:** Proposed method for isolating exosomes from raw milk.

Steps	IP Modified Method
Centrifugation to defat milk	2000 *g* at 4 °C, 10 min
Centrifugation to remove cells and cell debris	12,000 *g* at 4 °C, 40 min
Distilled water addition	1:1
Warming	37 °C, 10 min
Casein precipitation:	
Acidification	HCl 6N to adjust pH 4.6
Centrifugation	5000 *g* RT, 20 min
Freezing	−80 °C, overnight
Filtration	1.0, 0.45, 0.2 μm
Ultracentrifugation	100,000 *g* at 4 °C, 1 h
Pellet resuspension	0.1M PBS pH 7.4

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
