# Peer review of "MicroRNA Milk Exosomes: From Cellular Regulator to Genomic Marker"

_animals, 2020, doi:10.3390/ani10071126_

Round 1

Reviewer 1 Report

The paper of Cintio et al, entitled « Exosomes and miRNAs in milk: a cell-to-cell regulator and genomic phenotype » is a review of recent literature on breast milk extracellular vesicles. After briefly summarizing the field, the authors are focussing on methodological aspects of isolation and purification of exosomes from complex food like milk. They are presenting data on the cell-to-cell communication through exosome vehicles and proof of transfer of nucleic acids from these nanoparticles to the cytoplasm. At the end, they are bringing into perspective their work on Single nucleotide polymorphism as a means for selecting bovine strains resistant to mastitis.

The review is interesting, well-documented, and illustrated. It deserves publications however I am rising below some critics which might improve the paper. Leaving authors free to include or not.

Major modifications

1. Authors have adapted an illustration of Kalluri, R.; LeBleu (2020) Science.

This kind of representation can bring misunderstanding because ribonucleic acids are always displayed right at the center of the exosome. However, they can be at the surface or within lipoprotein complexes in the proximity of exosome plasma membrane. A mention of this possibility in the legend may help readers to avoid oversimplification (or you can underline that this is an artistic view).

2. Authors have not discussed data in contradiction with the hypothesis that the exosomes are miRNA shuttle hypothesis (Chevillet et al., 2014 Proc Natl Acad Sci USA. (2014) 111:14888–93. doi: 10.1073/pnas.1408301111). In fact, a word of caution may be included in that direction because due to methodological shortcomings (as stated by the authors), miRNA may be more contaminants of exosome purification process than a constituent of the exosome.

Author Response

Responses to reviewer 1

We would like to thank the reviewer for their valuable and constructive comments.  The queries raised (Q) have been responded to (R) below.

Q: Authors have adapted an illustration of Kalluri, R.; LeBleu (2020) Science. This kind of representation can bring misunderstanding because ribonucleic acids are always displayed right at the center of the exosome. However, they can be at the surface or within lipoprotein complexes in the proximity of exosome plasma membrane. A mention of this possibility in the legend may help readers to avoid oversimplification (or you can underline that this is an artistic view).

R: We have specified that Fig.1 is just a simplified representation (see legend).

Q: Authors have not discussed data in contradiction with the hypothesis that the exosomes are miRNA shuttle hypothesis (Chevillet et al., 2014 Proc Natl Acad Sci USA. (2014) 111:14888–93. doi: 10.1073/pnas.1408301111). In fact, a word of caution may be included in that direction because due to methodological shortcomings (as stated by the authors), miRNA may be more contaminants of exosome purification process than a constituent of the exosome.

R: As suggested we have underlined these different hypothesis (121-124 lines) and the reference was added [30].

Reviewer 2 Report

The objective of this review manuscript was to describe the possible application of EXO cargo as biomarkers in animal breeding programs. In general, the authors did a great job of explaining different aspects of Exo biology. However, a minimal discussion has been provided about how EXO cargo can be integrated with breeding programs (section 4, lines 407-457).

Minor revisions

Line 108- Authors argued that EXO-derived miRNAs are conserved across different species. Please provide a sentence describing whether or not the target of these miRNAs conserved across species

Author Response

Responses to reviewer 2

We would like to thank the reviewer for their valuable and constructive comments.  The queries raised (Q) have been responded to (R) below.

Q: The objective of this review manuscript was to describe the possible application of EXO cargo as biomarkers in animal breeding programs. In general, the authors did a great job of explaining different aspects of Exo biology. However, a minimal discussion has been provided about how EXO cargo can be integrated with breeding programs (section 4, lines 407-457).

R: Section 5 was completely re-written and Table 4 was deleted. Now paper is more finalized only to miRNAs in exosomes from milk that are now indicated in the text as EXO.

Q: Line 108- Authors argued that EXO-derived miRNAs are conserved across different species. Please provide a sentence describing whether or not the target of these miRNAs conserved across species

R: A sentence was added (lines 140-142).

Reviewer 3 Report

The title of the manuscript is not consistent with the content of the manuscript as most section of the manuscript the authors used to describe the methods related to EXO

The introduction is very weak and makes confusing rather than giving the audiences knowledge about the contents of the manuscript and related topics

The main contents about miRNAs roles are randomly picked and poorly organized, perhaps the authors should spend more time on it

The authors tried to bring some sentences regarding selections, but it is not clear how they are linked

The conclusion really makes the manuscript as a draft version.

I would things the authors should focus on the technical aspect of EXO and how they can use them as the phenotypes in the genetic or genomic selection, which could be much more interesting.

Comments

Line 12-13: please break the sentence or re-text as the sentence is hard to read.

You are also mixing the miRNAs and ncRNAs, note that we have different types of non-coding RNAs

Milk composition= mink compositions as you mentioned more than compositions in milk

Line 22: why you want to include the genome here, do you mean the DNA variants in the whole genome?

Line 23: ruminant to ruminant’s

The abstract is not well-organized. In fact, I could not recognize which type of paper (review or original) after reading the abstract. The authors mixed the miRNAs and EXO

What is the abbreviation of EXO, is it milk-derived exosomes or just exosomes? If the authors use for exosomes then it is not necessary, if it if for milk-derived exosomes then remove the milk in “milk EXO-derived miRNAs”.

Many sentences or phases are not needed in the abstract as such: “the potential use of miRNAs as a biomarker of disease resistance and metabolic resilience has recently received attention from the scientific community”.

“Several miRNA classes have been described as implicated in epigenetic control (miR-148a, miR-125b)” I am not sure miR-148a and miR-125b are considered as a class of miRNAs. Do they have some common role for all the ruminant species or just for bovine?

Line 37-38: instead of bringing the details issues in the abstract, I would expect some summary of the technical difficulty and some arguments about the roles of EXO.

How the authors want to use the DNA variants in miRNAs in the genome, the conclusion of abstract is far from what presented in the abstract.

Please reorganizing the abstract.

Line 48: some double space in the sentence

Line 50: don’t use the abbreviation at the beginning of a sentence

Line 54: released is for what subject?

Line 55: if you want to have an abbreviation for exosomes, then using them consistently

Line 58-60: it is not clear that the authors want to mention, please re-text

Line 92-93: Please insert a reference, since the % is arguable

From 102 -109: this is a very debated topic now.

Line 110 to 117:  Why authors place many sentences in the introductions in the role of milk?

I am not sure what the author's intent in the introduction? To get the audience in the topics or to list what they know about the topics.  Please give some outline information for the topics, what you want to inform the audience in the review papers: If you want to summarise the progress of EXO or if you want to a discussion about the limitation of methods.

Some paragraphs lacking the topic sentence, and some having just one sentence? Please combine and reconnect the information.

Tables 1 and 2 are not well-formatted, please following the guideline.

Line 149-151: I am not sure their statement could make some useful information for readers. If the choice depending on the starting sample, how the authors want to get the agreed protocol or methods as previously stated in the abstract

Line 193-194: The authors should give references for it

Line 218:  between 2012 and 2015 the authors should give the shorter time frame or just remove it

Line 365: you don’t need to make the text different here

Line 362-to end

 From the title, I expect much deeper about this section, but it is not what I want to see. As the authors spend all energy to describe the technical aspect of characterizing EXO and now, I have a feeling that the authors rush into summarise the information about the roles of miRNAs and EXO.

The mixing of information between the cows and humans, miRNAs, and EXO making the papers is hard to read.

The authors try to insert the information about the variants in miRNAs and selection which is less convinced and bringing away topics far from their own manuscript title.

Finally, I am not sure I am learning something about the paper except that I know many different methods for the isolation of EXO.

Line 363-365: I am not the fan of the sentence without meaning information, please tell us what you want to give scientific community learning about the topic in the paragraph

Line 363-371: The authors stating the paragraph with text as  "in human", and ending the paragraph as in mice, the paragraph has only three sentences which are not sure for which purpose?

Line 383: Why the authors wanted to bring “and mRNAs” here, and what if the sentence if for and what is the reference for it

Line 384-385: In which species

Line 385-386: In which species

Line 387: Where is the evidence?

Line 396: You should join this one with previous paragraph as you are giving an example for the immune function of miRNA and let-7a are among them

I don’t know which the headline “miRNA and selection” linking to the contents the section. Moreover, it is better you give the audience about what you intend to do in the section, giving the lists of ideas you want to cover in the section before starting to describe some details

I don’t know why you pick up some miRNAs for section, you should give a comprehensive summary instead of giving some random information

Line 429-436: You start with some idea about breed and

The conclusion is not the one for a review paper, you bring the last sentence without any connecting information about.

Author Response

Responses to reviewer 3

We would like to thank the reviewer for their valuable and constructive comments.  The queries raised (Q) have been responded to (R) below.

Q: Extensive editing of English language and style required

R: An English speaker checked English language

Q: The title of the manuscript is not consistent with the content of the manuscript as most section of the manuscript the authors used to describe the methods related to EXO

R: As suggested, we changed the title according to the changes made along the text.

Q: The introduction is very weak and makes confusing rather than giving the audiences knowledge about the contents of the manuscript and related topics

R: We changed part of the introduction, but we believe that a general overview on the features and role of the exosomes, EXO and their cargos are useful for the readers.

Q: The main contents about miRNAs roles are randomly picked and poorly organized, perhaps the authors should spend more time on it

R: According to other comments below, we have re-arranged the section about miRNAs (section 4).

Q: The authors tried to bring some sentences regarding selections, but it is not clear how they are linked

R: According to other comments below, we have re-arranged the discussion regarding selection (section 5).

Q: The conclusion really makes the manuscript as a draft version.

R: Conclusion was completely revised.

Q: I would things the authors should focus on the technical aspect of EXO and how they can use them as the phenotypes in the genetic or genomic selection, which could be much more interesting.

R: Our intention was to summarize some progress about EXO, which necessarily includes the methodological limitations. This is an integral part of the existing knowledge about exosomes. However, this section was limited to methodologies that are related to EXO (isolation of exosomes from milk). Section 5 on genomic selection was completely reviewed.

Q: Line 12-13: please break the sentence or re-text as the sentence is hard to read.

Q: You are also mixing the miRNAs and ncRNAs, note that we have different types of non-coding RNAs

R: From: Loudu Srijyothi, Saravanaraman Ponne, Talukdar Prathama, Cheemala Ashok and Sudhakar Baluchamy (October 10th 2018). Roles of Non-Coding RNAs in Transcriptional Regulation, Transcriptional and Post-transcriptional Regulation, Kais Ghedira, IntechOpen, DOI: 10.5772/intechopen.76125. “Though 80% of the human genome is transcribed into RNA, majority of RNA lacks protein coding potential and referred as “non-coding RNA” (ncRNA). ncRNAs are divided into two major groups based on an arbitrary threshold of 200 nucleotides (nt) namely short ncRNAs (sncRNA) and long ncRNAs (lncRNAs). sncRNAs include functional RNAs such as t-RNAs, r-RNAs and snRNAs which are involved in transcriptional and translational regulation. In addition to these conventional RNAs, short ncRNAs also include different regulatory RNAs such as microRNAs (miRNAs) [2, 3], small interfering RNAs (siRNAs) and P-element-induced wimpy testis (PIWI) interacting RNAs (piRNAs) [4], all of which regulate gene expression”

Said this, we changed ncRNA with short ncRNA (lines 25; 115) and changed simple summary.

Q: Milk composition= mink compositions as you mentioned more than compositions in milk

R: As suggested, we wrote compositions (line 26).

Q: Line 22: why you want to include the genome here, do you mean the DNA variants in the whole genome?

R: We have considered the comments and simple summary was re-written (lines 24-30).

Q: Line 23: ruminant to ruminant’s

R: Correction was done (line 51).

Q: The abstract is not well-organized. In fact, I could not recognize which type of paper (review or original) after reading the abstract. The authors mixed the miRNAs and EXO

Q: What is the abbreviation of EXO, is it milk-derived exosomes or just exosomes? If the authors use for exosomes then it is not necessary, if it if for milk-derived exosomes then remove the milk in “milk EXO-derived miRNAs”.

R: We have now clarified that EXO refer to milk-derived exosomes and we modified along the text.

Q: Many sentences or phases are not needed in the abstract as such: “the potential use of miRNAs as a biomarker of disease resistance and metabolic resilience has recently received attention from the scientific community”.

R: Sentences were deleted.

Q: “Several miRNA classes have been described as implicated in epigenetic control (miR-148a, miR-125b)” I am not sure miR-148a and miR-125b are considered as a class of miRNAs. Do they have some common role for all the ruminant species or just for bovine?

R: There was a misunderstanding. miR-148a and miR-125b do not form a class together. miRNAs comprise one of the more abundant classes of gene regulatory molecules in multicellular organism. To avoid any misunderstanding ‘classes’ was deleted.

Q: Line 37-38: instead of bringing the details issues in the abstract, I would expect some summary of the technical difficulty and some arguments about the roles of EXO.

Q: How the authors want to use the DNA variants in miRNAs in the genome, the conclusion of abstract is far from what presented in the abstract.

Q: Please reorganizing the abstract.

R: We have considered the comments and abstract was re-written (lines 51-62).

Q: Line 48: some double space in the sentence

R: Space was corrected

Q: Line 50: don’t use the abbreviation at the beginning of a sentence

R: Abbreviation was deleted

Q: Line 54: released is for what subject?

R: The subject is the relative ‘that’ (‘intraluminal vesicles’), ‘are’ is added ‘are released’ (line 74)

Q: Line 55: if you want to have an abbreviation for exosomes, then using them consistently

R: Here exosomes are generally considered. According to your previous comment, we used EXO for milk-derived exosomes.

Q: Line 58-60: it is not clear that the authors want to mention, please re-text

R: Sentence was re-wrote, abstract was re-written.

Q: Line 92-93: Please insert a reference, since the % is arguable

R: References are added (line 116) [21,22].

Q: From 102 -109: this is a very debated topic now.

R: Recent paper by Manca et al (2018) clearly demonstrated the intestinal uptake of bovine EXO in mice. We provided ref. for that.  [49] (line 146)

Q: Line 110 to 117:  Why authors place many sentences in the introductions in the role of milk?

R: We re-wrote the sentence. Introduction was partly re-written (lines 130-137).

Q: I am not sure what the author's intent in the introduction? To get the audience in the topics or to list what they know about the topics.  Please give some outline information for the topics, what you want to inform the audience in the review papers: If you want to summarise the progress of EXO or if you want to a discussion about the limitation of methods.

R: Our intention is to summarize some progress about EXO, which necessarily includes the methodological limitations. This is an integral part of the existing knowledge about exosomes. For this reasons we deleted methodologies that are not related to EXO (isolation of exosomes from milk).

Q: Some paragraphs lacking the topic sentence, and some having just one sentence? Please combine and reconnect the information.

R: We re-wrote this paragraph

Q: Tables 1 and 2 are not well-formatted, please following the guideline.

R: Tables were formatted according to guidelines.

Q: Line 149-151: I am not sure their statement could make some useful information for readers. If the choice depending on the starting sample, how the authors want to get the agreed protocol or methods as previously stated in the abstract

R: There was a misleading and we re-wrote the sentence, hoping that is clearer now  (lines 199-201). We would like to suggest an agreed protocol for milk.

Q: Line 193-194: The authors should give references for it

R: References are in table numbered in brackets.

Q: Line 218:  between 2012 and 2015 the authors should give the shorter time frame or just remove it

R: We deleted “between 2012 and 2015” (line 271).

Q: Line 365: you don’t need to make the text different here

R: Of course, it was a misprint!

Line 362-to end

Q: From the title, I expect much deeper about this section, but it is not what I want to see. As the authors spend all energy to describe the technical aspect of characterizing EXO and now, I have a feeling that the authors rush into summarise the information about the roles of miRNAs and EXO.

R: We guess what you mean. However, we focused the review on exosomes in milk and miRNAs in EXO in dairy ruminants, especially cows. We have reduced methodological section focusing just on method for exosome isolation in milk.  We collected existing information on this topic to attract the reader on the potential that this new knowledge can provide to animal science.

Q: The mixing of information between the cows and humans, miRNAs, and EXO making the papers is hard to read.

R: Yes, of course. Several reviews talked about human, animal and rodents, especially when information is limited for one species (i.e. cow). Studies of miRNAs in exosomes on other species are related to pathological conditions as breast cancer and we tried to extrapolate the information useful to mammary glands biology of cows.

Q: The authors try to insert the information about the variants in miRNAs and selection which is less convinced and bringing away topics far from their own manuscript title.

R:  Section 5 was re-written

Q: Finally, I am not sure I am learning something about the paper except that I know many different methods for the isolation of EXO.

R: We hope that after the revision of the manuscript you will be able to learn something more about exosome in milk.

Q: Line 363-365: I am not the fan of the sentence without meaning information, please tell us what you want to give scientific community learning about the topic in the paragraph

R: The sentence for modified and the text moved to lines 459-464 with further information.

Q: Line 363-371: The authors stating the paragraph with text as  "in human", and ending the paragraph as in mice, the paragraph has only three sentences which are not sure for which purpose?

R: We agreed and changed the sentence (lines 442-448)

Q: Line 383: Why the authors wanted to bring “and mRNAs” here, and what if the sentence if for and what is the reference for it

R: Sorry, but we are not sure to understand what you mean. We have deleted the sentence with ‘mRNA’.

Q: Line 384-385: In which species

R: Species are reported now (lines 468; 470)

Q: Line 385-386: In which species

R: Bovine was added (line 456)

Q: Line 387: Where is the evidence?

R: This is the conclusive sentence reported in references [131-133] (lines 467-470).

Q: Line 396: You should join this one with previous paragraph as you are giving an example for the immune function of miRNA and let-7a are among them

R: We agreed and arranged the section.

Q: I don’t know which the headline “miRNA and selection” linking to the contents the section. Moreover, it is better you give the audience about what you intend to do in the section, giving the lists of ideas you want to cover in the section before starting to describe some details

Q: I don’t know why you pick up some miRNAs for section, you should give a comprehensive summary instead of giving some random information

Q: Line 429-436: You start with some idea about breed and

R: Section 5 was completely re-written (lines 485-515) and Table 4 was deleted.

Q: The conclusion is not the one for a review paper, you bring the last sentence without any connecting information about.

R: Conclusion was completely re-written.

Round 2

Reviewer 3 Report

In the summary:

as genomic phenotype: might change to as a phenotype for genomic studies

In the abstract:

Line 19: Short noncoding RNA: might be better used as microRNAs as you focus on the miRNA only

Line 19: You need to define the abbreviation for miRNAs

Line 21: “Although studies” you might to “Although many studies”

Line 24: are they also involved in the cellular signaling pathways?

Line 25: “the isolation of EXO still represents a challenge” I would suggest that to change to “ the studies of miRNAs EXO is still challenging due to the difficulty of isolation of EXO” to better fit with the context in the previous sentence.

Line 28: Might be you don’t need to specify as dairy ruminants, as they can be used in other livestock species

Line 86 and other place: exosome should be used in abbreviation

Table 2: You need to use the author's name here and add the reference numbers after.

Line 363: Quite a lot of  reviewing papers regarding the roles of miRNAs such as (Wang et al. 2013; Do & Ibeagha-Awemu 2017; Van Hese et al. 2020), which might be good if you can give the readers some information regarding, in comparison with the studies focusing on miRNAs EXO.

Line 379: Some other aspects of miRNAs such as they are participating in various signaling pathways (Do et al. 2017b), or the interactions among MiRNAs in regulation of many milk traits (Do et al. 2017a; Ammah et al. 2018) and they are epigenetics regulators (Melnik & Schmitz 2017a, b) might be important to enrich the miRNAs sections

  1. Ammah A.A., Do D.N., Bissonnette N., Gévry N. & Ibeagha-Awemu E.M. (2018) Co-expression network analysis identifies miRNA–mRNA networks potentially regulating milk traits and blood metabolites. International journal of molecular sciences 19, 2500.
  2. Do D.N., Dudemaine P.-L., Li R. & Ibeagha-Awemu E.M. (2017a) Co-expression network and pathway analyses reveal important modules of miRNAs regulating milk yield and component traits. International journal of molecular sciences 18, 1560.
  3. Do D.N. & Ibeagha-Awemu E.M. (2017) Non-coding RNA roles in ruminant mammary gland development and lactation. Current Topics in Lactation, 55.
  4. Do D.N., Li R., Dudemaine P.-L. & Ibeagha-Awemu E.M. (2017b) MicroRNA roles in signalling during lactation: an insight from differential expression, time course and pathway analyses of deep sequence data. Scientific reports 7, 1-19.
  5. Melnik B.C. & Schmitz G. (2017a) MicroRNAs: Milk's epigenetic regulators. Best Practice & Research Clinical Endocrinology & Metabolism 31, 427-42.
  6. Melnik B.C. & Schmitz G. (2017b) Milk’s role as an epigenetic regulator in health and disease. Diseases 5, 12.
  7. Van Hese I., Goossens K., Vandaele L. & Opsomer G. (2020) Invited review: MicroRNAs in bovine colostrum—Focus on their origin and potential health benefits for the calf. Journal of dairy science 103, 1-15.
  8. Wang X., Gu Z. & Jiang H. (2013) MicroRNAs in farm animals. Animal: an international journal of animal bioscience 7, 1567.

Author Response

Responses to reviewer 3

We would like to thank the reviewer for their valuable and constructive comments.  The queries raised (Q) have been responded to (R) below.

In the summary:

Q: as genomic phenotype: might change to as a phenotype for genomic studies

R: According to suggestion, words were changed (line 17)

In the abstract:

Q: Line 19: Short noncoding RNA: might be better used as microRNAs as you focus on the miRNA only

R: According to suggestion, Short noncoding RNA was changed with microRNAs (line 19)

Q: Line 19: You need to define the abbreviation for miRNAs

R: miRNAs abbreviation was defined (line 19)

Q: Line 21: “Although studies” you might to “Although many studies”

R: ‘Many’ was added.

Q: Line 24: are they also involved in the cellular signaling pathways?

R: Certainly, the sentence was modified (lines 24-26).

Q: Line 25: “the isolation of EXO still represents a challenge” I would suggest that to change to “ the studies of miRNAs EXO is still challenging due to the difficulty of isolation of EXO” to better fit with the context in the previous sentence.

R: The sentence was changed as required (lines 26-28).

Q: Line 28: Might be you don’t need to specify as dairy ruminants, as they can be used in other livestock species

R: Word was changed as required (line 31)

Q: Line 86 and other place: exosome should be used in abbreviation

 R: As reported in the Simple summary, Abstract and in lines 48-49, along the paper EXO refers to exosomes in milk, while exosome/s refers to any other source of exosomes.

Q: Table 2: You need to use the author's name here and add the reference numbers after.

R: Author’s name was added in Table 2.

Q: Line 363: Quite a lot of  reviewing papers regarding the roles of miRNAs such as (Wang et al. 2013; Do & Ibeagha-Awemu 2017; Van Hese et al. 2020), which might be good if you can give the readers some information regarding, in comparison with the studies focusing on miRNAs EXO.

Q: Line 379: Some other aspects of miRNAs such as they are participating in various signaling pathways (Do et al. 2017b), or the interactions among MiRNAs in regulation of many milk traits (Do et al. 2017a; Ammah et al. 2018) and they are epigenetics regulators (Melnik & Schmitz 2017a, b) might be important to enrich the miRNAs sections.

R: Thank you for suggestion, we added a sentence reporting some of the references [144,145,146] (lines 367-369).

Melnik & Schmitz 2017a was added [136] to support epigenetic regulation in line 347.

Van Hese et al., was already cited [57].